# A Simplified PET/CT Measurement Routine with Excellent Diagnostic Accuracy for the Diagnosis of Giant Cell Arteritis

**DOI:** 10.3390/diagnostics12030728

**Published:** 2022-03-17

**Authors:** Stephan Imfeld, Delia Scherrer, Noemi Mensch, Markus Aschwanden, Daniel Staub, Christoph T. Berger, Thomas Daikeler, Christof Rottenburger

**Affiliations:** 1Department of Angiology, University Hospital Basel, 4031 Basel, Switzerland; stephan.imfeld@usb.ch (S.I.); markus.aschwanden@usb.ch (M.A.); daniel.staub@usb.ch (D.S.); 2Department of Rheumatology, University Hospital Basel, 4031 Basel, Switzerland; delia.scherrer@hotmail.com (D.S.); noemi.mensch@stud.unibas.ch (N.M.); 3University Center for Immunology, University Hospital Basel, 4031 Basel, Switzerland; christoph.berger@usb.ch; 4Translational Immunology, Department of Clinical Research, University of Basel, 4031 Basel, Switzerland; 5Division of Nuclear Medicine, University Hospital Basel, 4031 Basel, Switzerland; christof.rottenburger@usb.ch

**Keywords:** giant cell arteritis, large-vessel-vasculitis, PET/CT, imaging, diagnosis, SUV, validation

## Abstract

We previously proposed standard uptake value (SUV) ratio-based cut-off values for [^18^F] fluorodeoxyglucose-positron emission tomography/computed tomography (PET/CT) for diagnosing giant cell arteritis (GCA) with high diagnostic accuracy. Here we confirm our findings in an independent cohort and report a simplified procedure for using a SUV ratio to diagnose LV-GCA. Patients with suspected GCA were consecutively included. The ‘peak SUV ratio’ was defined in a two-step approach. First, the vessel with the visually brightest radiotracer uptake in the supra-aortic (SA) and in the aorto-iliofemoral (AIF) region was identified. Here, the maximum SUV of the vessel was measured and divided by the mean SUV of the liver (SUV_ratio_). A ratio >1.0 in the SA or >1.3 in the AIF region was scored as vasculitis. The diagnostic accuracy, sensitivity, and specificity of the ‘peak SUV ratio’ in the SA and AIF region was assessed. From 2015 to 2019, 50 patients (24 female, median age 71 years) with suspicion of GCA were included, 28 patients with GCA and 22 patients with exclusion of GCA. Peak SUV had an AUC of 0.91, a sensitivity of 0.89, and a specificity of 0.73 for diagnosing GCA. Peak SUV accuracy of the AIF arteries was lower (AUC 0.81) than of the SA arteries (AUC 0.95). Our SUV ratio cut-off values for diagnosing GCA are consistently valid, also when applied in a time-efficient clinical procedure focusing on the peak SUV ratio. The diagnostic performance of PET/CT in this validation cohort was even higher, compared to the inception cohort (AUC of 0.83).

## 1. Introduction

Diagnosis of large-vessel giant cell arteritis (LV-GCA) with predominant extracranial manifestation remains challenging. Imaging plays an important role in the diagnosis of GCA. Ultrasound of the temporal and the axillar artery is often used, but may be false negative in case of vasculitic involvement of the larger arteries only. For these cases, and also for patients lacking cranial symptoms, PET/CT has a potentially higher diagnostic accuracy than ultrasound. Moreover, both techniques may be used complementary, enhancing sensitivity for the diagnosis of GCA [1]. [^18^F] Fluorodeoxyglucose-positron emission tomography/computed tomography (PET/CT) is well established and has been integrated in the current guidelines for diagnosing LV-GCA [2]. However, clear recommendations for how to interpret PET/CT are lacking [3]. The simplest approach of qualitative scoring is based on visual uptake pattern [4,5] and, therefore, operator dependent. A semi-standardized approach visually compares vessel FDG-uptake with liver uptake according to a four-grade scale [6]. Direct measurements of vessel standard uptake value (SUV) or SUV to background ratios such as the vessel-to-liver SUV ratio, have been proposed to improve the quantification of tracer uptake [7]. This approach outperforms visual scoring, reduces interobserver variability, and enables comparison of ratios, enabling disease activity monitoring over time [8]. To allow comparison across studies, the definition of reliable, diagnostic SUV cut-off ratios is critical for the clinical application of a SUV-ratio-based approach. In our previous study (‘inception cohort’), quantitative SUV-ratio-based scoring out-performed visual scoring (specificity 0.86 vs. 0.77, with similar sensitivity of 0.72 vs. 0.75) [8].

The aim of this study was to expand on these findings and to facilitate the clinical application of our approach. We applied the previously proposed cut-off values to an independent confirmation cohort and simplified the SUV ratio determination by focusing only on the supra-aortic (SA) and aorto-iliofemoral (AIF) region with the visually highest FDG uptake.

## 2. Material and Methods

Patients presenting at the University Hospital Basel that underwent a PET/CT scan for suspected giant cell arteritis were included in our local ethics committee (EKNZ 239/09) approved prospective cohort of GCA patients (BARK).

The study was performed in accordance with the Declaration of Helsinki. The diagnosis of GCA was confirmed either (i) by positive temporal artery (TA) biopsy, (ii) if the 1990 ACR criteria were fulfilled, or (iii) if at least 2/5 ACR criteria were fulfilled, combined with typical ‘vasculitic’ findings in ultrasound or signs of vasculitis in other imaging modalities [9,10,11]. Clinical, laboratory, and treatment data were extracted from our local, prospective GCA cohort and from the electronic hospital charts.

### 2.1. PET/CT Scan Acquisition

All PET/CT scans were performed at the University Hospital Basel on a Siemens Biograph PET/CT mCT128 scanner (Siemens Healthcare, Erlangen, Germany). Fasting for at least 6 h before injection of the radiotracer was required. Scans were started 1 h after intravenous injection of 5 MBq ^18^F-FDG/kg body weight at median glycaemia levels of 5.4 mmol/L (interquartile range 4.9–5.9). A native CT scan of the skull was performed to enable attenuation correction as well as morphological correlation in a supine position with the upper extremities beside the body. The scans were performed with 120 kV, using automatic expose with a reference setting of 70 mAs. Then, the PET emission scan was performed with one bed position of 10 min in 3D mode. Next, a whole body CT scan (skull base to tights with arms beside the skull) was acquired at 120 kV with a reference setting of 50 mAs. The subsequent PET scan of the latter region was acquired with 90 s per bed position. Image reconstruction, using time of fight technique, was performed iteratively using five iterations with 21 subsets. The reconstruction parameters were: Gauss filter (FWHM 2 mm) and a 400 × 400 matrix (skull), Gauss filter (5 mm), and a 200 × 200 matrix (whole body). To avoid interference with the attenuation correction, which is necessary for the reconstruction of the PET scans, we performed the CT scan without contrast media.

### 2.2. PET/CT Scan Analysis

PET/CT scans were analysed with Siemens SyngoVia software by a nuclear medicine expert (CR), blinded for all patient data besides the PET/CT scan. Vessels in the SA (A. vertebralis, carotis, subclavia, and axillaris) and the AIF (Aorta thoracica and abdominalis, A. iliaca and femoralis communis) were assessed visually. For each of the two regions, the vessel with the highest visual FDG-uptake was used to measure SUV and calculate the SUV vessel–liver ratio (i.e., two values per patient). This step was introduced to facilitate and speed-up the SUV ratio determination, making it suitable for daily clinical practice. Mean SUV of the liver was measured as reference value, and vessel-to-liver ratios (SUV_ratios_) were calculated (Figure 1). A SUV_ratio_ >1.0 in the supra-aortic or >1.3 in the AIF region in PET/CT was scored as vasculitis, as previously defined [8].

### 2.3. Statistics

Continuous variables were analysed using the Mann–Whitney U test and were expressed as medians and interquartile ranges. Categorical variables were analysed with the Fisher’s exact test. *p*-values < 0.05 were considered significant. Statistical analysis was performed using GraphPad Prism 8.3.0 (Graphpad Software, San Diego, CA, USA).

## 3. Results

### 3.1. Patient Characteristics

Fifty patients with a mean age of 70 years (range 54–88) were included between August 2015 and January 2019. In 28 patients (56%), GCA was confirmed by final diagnosis. Twenty-one out of the 28 GCA patients fulfilled three or more of the ACR criteria. Six patients fulfilled 2/5 ACR criteria and had a clearly positive TA biopsy (*n* = 1) or vasculitic findings in the TA ultrasound (*n* = 5). One patient was included during relapse of giant cell arteritis. The 22 patients (44%) in which GCA was ruled out served as the control group. Patients of the control group were diagnosed with polymyalgia rheumatica (*n* = 8), anterior ischemic optic neuropathy (*n* = 4), inflammatory syndromes (*n* = 3), infection (*n* = 2), papilloedema (*n* = 1), cholangiocellular carcinoma (*n* = 1), myelodysplastic syndromes (*n* = 1), pulmonary embolism (*n* = 1), and diverticulitis (*n* = 1).

The GCA patients had a higher median erythrocyte sedimentation rate (ESR) than the controls but were comparable in their other characteristics (Table 1). Fifteen GCA patients were glucocorticoid-naïve at the time of PET/CT. Three patients received long-term corticosteroid therapy, two because of previous diagnosis of polymyalgia rheumatica (*n* = 5, 10 mg), and one because of previous GCA diagnosis (2.5 mg). The patient with GCA was undergoing PET/CT for suspicion of relapse. The remaining 10 patients received higher cumulative corticosteroid doses for a short time before PET/CT (Table 1). Eleven control patients were glucocorticoid-naïve at the time of PET/CT. One patient received long-term corticosteroid therapy (10–20 mg), due to inflammation of unknown origin, which later was diagnosed as myelo-dysplastic syndrome (MDS). The remaining 10 patients received higher cumulative corticosteroid doses for a short term before PET/CT.

### 3.2. PET/CT Results

Overall, PET/CT had a sensitivity of 0.89 and a specificity of 0.73 for GCA using the peak SUV–vessel/liver ratio at the two investigated vessel regions combined. Considering the two regions separately, the analysis of the SA region resulted in a sensitivity of 0.82 and a specificity of 0.91, whereas the AIF region resulted in lower values with a sensitivity of 0.75 and a specificity of 0.73. ROC analysis showed an area under the curve (AUC) of 0.91 (±0.04 standard error (SE)) for the overall analysis, 0.95 (±0.03 SE) for the SA region, and 0.81 (±0.06 SE) for the AIF region. The vessel that most frequently had the highest SUV_ratio_ values of the SA vessels was the subclavian artery (peak SUV ratio in 22/28 cases). Three out of the 28 patients with GCA showed isolated involvement of the cranial vessels (vertebral arteries) only, with SUV_ratios_ of 1.4 in one and 1.5 in the remaining two patients. In the AIF region, the peak SUV_ratio_ values most frequently appeared within the abdominal aorta (22/28 cases). In 19/22 patients in the control group, atherosclerotic lesions were detectable in the low dose CT. Overall, the diagnostic accuracy of PET/CT using the simplified protocol cohort was superior to our inception cohort [8] for both vessel regions (Figure 2).

## 4. Discussion

Herein, we confirmed the diagnostic accuracy of our previously established PET/CT SUV_ratio_ cut-off criteria (>1.0 for the SA region and >1.3 for the AIF region) for LV-GCA with a sensitivity of 0.89 and a specificity of 0.73. In ROC analysis, the AUC of 0.91 surpassed the value of 0.83 in the inception cohort, despite the fact that we simplified the PET/CT analysis steps by measuring only the SUV_ratios_ of the brightest visual uptake signal in a vessel of the SA and AIF region, each. This adaption to the previously published procedure is by far more time-efficient and, therefore, is suitable for application in daily routine, which is a relevant advantage compared to more time-consuming quantification methods [12]. Importantly, this is achieved without loss of diagnostic accuracy but still adding a higher grade of standardisation. Sensitivity of PET/CT in this validation cohort was even higher than in the inception cohort (0.89 vs. 0.72). Various reasons may have contributed to this better performance in the validation cohort including a shorter median duration of prednisone intake of GCA patients before PET/CT in GCA patients (5 vs. 6 days) and higher systemic inflammatory markers in the GCA patients compared to the inception cohort. Unfortunately, the number of GCA patients receiving high dose steroid treatment before PET/CT was too low to allow a separate ROC subgroup analysis. We previously established that the duration of prednisone intake before imaging strongly influences the intensity of tracer uptake in PET/CT and therefore its diagnostic accuracy [8]. According to this, imaging should best be performed within the first 10 days after initiation of treatment. In addition, Walter et al. showed that the levels of acute disease activity markers in serum correlate positively with the uptake intensity in PET/CT [5]. In both, the validation and inception cohort, diagnostic accuracy of PET/CT was higher in the SA region (AUC 0.95 respective 0.83) than in the AIF region (AUC 0.81 respective 0.69). Combining the results from both vessel regions resulted in the identification of two more GCA patients, at the cost of four more patients being false positive compared to the SA region alone. Therefore, in cases of aortic and iliofemoral vasculitic PET/CT findings, alternative explanations for high uptake, such as atherosclerotic burden, or alternative diagnoses, should be thoroughly evaluated [13]. However, the majority of our patients in the control group had detectable atherosclerotic lesions visible in the CT, therefore, a significant limitation of the SUV_ratio_ specificity by atherosclerotic lesions can be excluded.

The heterogeneous patient collective may be a limitation to our study. However, the study population represents a clinical ‘real world’ scenario of patients with suspected GCA, including LV-GCA and cranial GCA, each with and without steroid therapy, as well as conditions that can mimic GCA. Despite introducing some heterogeneity, we consider this to be a more realistic scenario, compared to direct comparisons to cohorts recruited from oncological, inflammatory, or normal patient cohorts [7,14]. Another limitation is the moderate sample size, which could result in less robust estimations. Although PET/CT analysis was performed by one experienced reader only, the simplified procedure of quantifying PET/CT with two measurements in the SA and the AIF region does not seem to degrade the diagnostic value of the analysis, but considerably increases speed and practicability in clinical routine applications.

In conclusion, this data confirms the validity of the SUV_ratio_ based PET/CT analysis for diagnosing GCA, especially when focusing on the SA vessels using a cut-off >1. The diagnostic accuracy of the AIF vessels (cut-off >1.3) is lower and should be interpreted with caution.

## Figures and Tables

**Figure 1 diagnostics-12-00728-f001:**
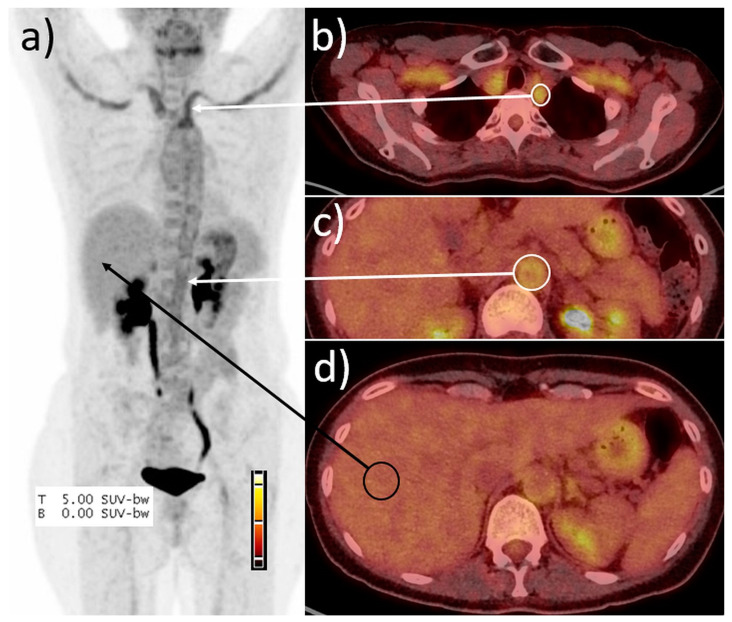
Example of the peak SUV ratio determination. (**a**) Maximum intensity projection, (**b**–**d**) fused PET/CT images. Calculation of the SUV_ratio_ exemplified in patient #15. SUV_max_ in the left subclavian artery (white circle, **b**) was 3.7, in the abdominal aorta (**c**) 2.1, and in the right liver lobe (black circle) SUV_mean_ was 1.6, resulting in a SUV_ratio_ of 2.2 in maximum. In consequence, the scan was scored positive for vasculitis.

**Figure 2 diagnostics-12-00728-f002:**
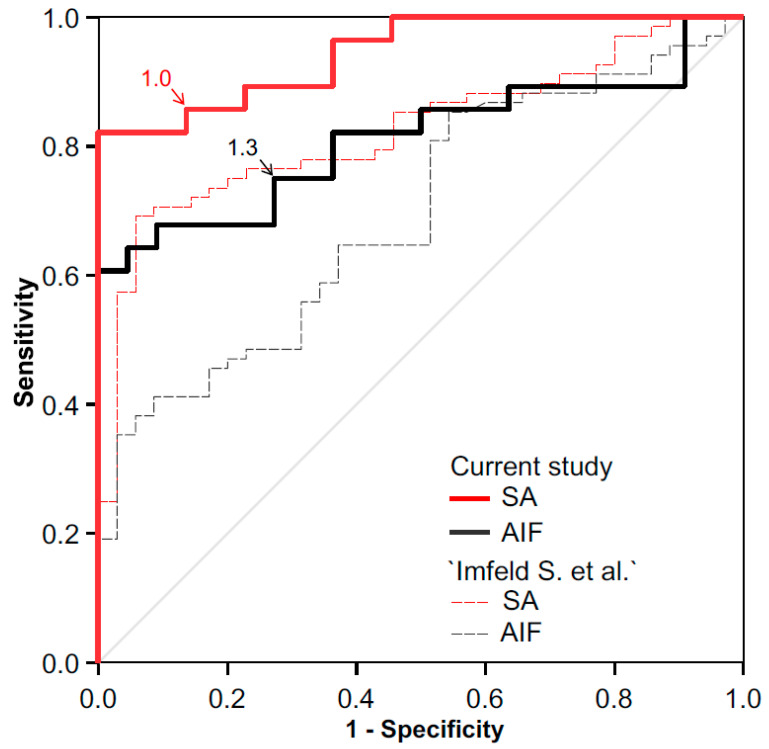
Receiver operating curves (ROC) of SUV vessel_max_/liver_mean_ ratios for the supra-aortic (SA) and the aorto-iliofemoral (AIF) region. For comparison, ROC of the inception cohort Imfeld S. et al. [8] are displayed as thin dotted lines.

**Table 1 diagnostics-12-00728-t001:** Patients characteristics: Data are expressed as percentages (numbers) or median (interquartile range). GCA = giant cell arteritis.

	GCA (*n* = 28)	Controls (*n* = 22)	*p*-Value
Female % (*n*)	54% (15)	41% (9)	0.56
Median Age in years (IQR)	73 (66–77)	68 (61–78)	0.31
Amaurosis fugax, loss of vision	36% (10)	45% (10)	0.57
Jaw claudication	36% (10)	14% (3)	0.11
New onset headache	61% (17)	64% (14)	>0.99
Scalp tenderness	39% (11)	27% (6)	0.34
Proximal muscle pain	50% (14)	45% (10)	>0.99
Shoulder pain	39% (11)	41% (9)	>0.99
Hip pain	29% (8)	41% (9)	0.55
Fever	11% (3)	32% (7)	0.15
Weight loss	29% (8)	32% (7)	>0.99
Night sweat	29% (8)	32% (7)	>0.99
Erythrocyte sedimentation rate (mm/h)	80 (46–90)	47 (23–64)	0.01
C-reactive protein (mg/L)	66 (26–111)	49 (14–122)	0.36
Corticosteroid-naïve	54% (15)	50% (11)	>0.99
Corticosteroid therapy: Cumulative dose (mg)	350 (243–600)	590 (285–1613)	0.45
Corticosteroid therapy: Duration of intake (days)	5 (3–8)	4 (3–6)	0.56

## Data Availability

Data available on request due to privacy restrictions. The data presented in this study are available on request from the corresponding author.

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
