# Peer review of "A Simplified PET/CT Measurement Routine with Excellent Diagnostic Accuracy for the Diagnosis of Giant Cell Arteritis"

_diagnostics, 2022, doi:10.3390/diagnostics12030728_

Round 1

Reviewer 1 Report

Imfeld et al report a validation study on using standard uptake values of vessel inflammation upon PET/CT normalized to the liver SUV, creating two ratios, one based on the values from the subclavian artery and one from the iliacal artery. This study is a well performed validation of their findings published in 2018 in Eur Heart J and the described SUV ratio could be applicable in routine PET/CT analyses, especially for vasculitis clinics. 
The manuscript is publishable as it is, I only have one question: 
- how many patients had diabetes? 

Author Response

Reviewer 1

Point 1: Imfeld et al report a validation study on using standard uptake values of vessel inflammation upon PET/CT normalized to the liver SUV, creating two ratios, one based on the values from the subclavian artery and one from the iliacal artery. This study is a well performed validation of their findings published in 2018 in Eur Heart J and the described SUV ratio could be applicable in routine PET/CT analyses, especially for vasculitis clinics. 
The manuscript is publishable as it is, I only have one question: 
- how many patients had diabetes? 

> Point 1: Thank you for your kind evaluation of the manuscript.  Two of the patients in the control group and one patient in the GCA group had diabetes.

Reviewer 2 Report

Imfeld et al. validated the use of SUV ratio in FDG-PET to diagnose GCA, and they concluded that its diagnostic performance was high. Because the use of PET-CT in large vessel vasculitis is increasing, simplified procedure to evaluate SUV would be important in daily practice. On the other hand, there exist several concerns in this study. 

1. It is unclear whether GCA with cranial lesion alone was included in this study. Were SUV ratios high in such patients? 

2. The most difficult differential diagnosis is atherosclerosis, and it is not difficult to diagnose GCA using CT/MRI in patients without atherosclerosis. The control group recruited in this study does not contain populations which possess arterial lesions mimicking GCA. 

My concern is that the atherosclerotic lesions sometimes show intense uptake of FDG, and therefore  simplified procedure used in this article might result in false positive in such patients. It would be better to show the data of the simplified procedure in patients with atherosclerosis to confirm their conclusions. 

3. As the authors mentioned in Discussion, the patients included in this study is heterogeneous. Because the uptake of FDG decreases after the initiation of treatments, it might be reasonable to use different cut-off among patients with and without treatment. The SUV in inactive patients or treated patients could become less than SUV in liver. Therefore, it would be informative to present additional ROC in patients with and without treatments to show whether there does not exist difference among these conditions. 

4. If the presence of treatments did not change the cut-off value, it is informative to discuss the timing of PET-CT. How long does the simplified procedure remain useful after treatments? 

5. There exist several modalities to detect arterial inflammation, and PET-CT is not usually an initial approach. It is informative to discuss how these modalities should be applied for the diagnosis of GCA. 

Author Response

Reviewer 2

Imfeld et al. validated the use of SUV ratio in FDG-PET to diagnose GCA, and they concluded that its diagnostic performance was high. Because the use of PET-CT in large vessel vasculitis is increasing, simplified procedure to evaluate SUV would be important in daily practice. On the other hand, there exist several concerns in this study. 

Point 1: It is unclear whether GCA with cranial lesion alone was included in this study. Were SUV ratios high in such patients?

>Point 1: Three out of 28 patients showed cranial involvement only, the SUV ratios were once 1.4 and twice 1.5, all measured in the vertebral artery and clearly above the threshold value.

Point 2: The most difficult differential diagnosis is atherosclerosis, and it is not difficult to diagnose GCA using CT/MRI in patients without atherosclerosis. The control group recruited in this study does not contain populations which possess arterial lesions mimicking GCA. 

My concern is that the atherosclerotic lesions sometimes show intense uptake of FDG, and therefore simplified procedure used in this article might result in false positive in such patients. It would be better to show the data of the simplified procedure in patients with atherosclerosis to confirm their conclusions. 

>Point 2: Thank you for this important comment. Indeed, arteriosclerosis is a potential confounder of vasculitis. Moreover, arteriosclerosis is prevalent in elderly patients. We aimed to simulate the clinical scenarios by including control patients that presented with suspicion of GCA. This is the same approach used by the DCVAS study aiming at establishing diagnostic and classification criteria for the primary vasculitides (https://research.ndorms.ox.ac.uk/public/dcvas/index.php). However, 19 of the control patients showed arteriosclerotic lesions in the CT scan (calcifications). Despite this, diagnostic accuracy of our scoring method was higher than in the inception cohort. Thus, arteriosclerosis may not be a major confounder for our simplified method.

Point 3:  As the authors mentioned in Discussion, the patients included in this study is heterogeneous. Because the uptake of FDG decreases after the initiation of treatments, it might be reasonable to use different cut-off among patients with and without treatment. The SUV in inactive patients or treated patients could become less than SUV in liver. Therefore, it would be informative to present additional ROC in patients with and without treatments to show whether there does not exist difference among these conditions.

> Point 3: Although sensitivity of PET/CT decreases after initiation of steroid treatment, vasculitic vessel SUV uptake may be visible as long as 3-6 months after initiation of therapy in PET/CT [1]. We previously showed a slightly lower sensitivity of PET/CT for GCA in patients having more than 10 days of prednisone treatment before PET/CT [2]. However, a lower cut-off for GCA would most likely reduce specificity of PET/CT for GCA. We thus would not propose different cut off values adapted to prednisone dose and duration. Clearly, imaging should be best performed within the first 10 days after initiation of treatment. In the current cohort, only one patient had high dose steroid treatment for more than 10 days before PET/CT, and only 5 patients for more than 5 days, which unfortunately precludes a separate ROC analysis for this subgroup.

Point 4: If the presence of treatments did not change the cut-off value, it is informative to discuss the timing of PET-CT. How long does the simplified procedure remain useful after treatments? 

> Point 4: We think the simplified method has the same limitations as our previously described method [2]. Please see our response above.

Point 5: There exist several modalities to detect arterial inflammation, and PET-CT is not usually an initial approach. It is informative to discuss how these modalities should be applied for the diagnosis of GCA. 

>Point 5: Indeed, different techniques may be used for diagnosing GCA and each has its advantages and disadvantages. We now added the sentence in the introduction: ‘Imaging plays an important role in the diagnosis of GCA. Ultrasound of the temporal and the axillar artery is often used, but may be false negative in case of vasculitic involvement of the larger arteries only. For these cases, and also for patients lacking cranial symptoms PET/CT has a potentially higher diagnostic accuracy than US. Moreover, both techniques may be used complementary enhancing sensitivity for the diagnosis of GCA [3].

1) Blockmans D, Stroobants S, Maes A, Mortelmans L. Positron emission tomography in giant cell arteritis and polymyalgia rheumatica: evidence for inflammation of the aortic arch. Am J Med. 2000; 108:246-9.

2)  Imfeld S, Rottenburger C, Schegk E, Aschwanden M, Juengling F, Staub D, Recher M, Kyburz D, Berger CT, Daikeler T. [18F]FDG positron emission tomography in patients presenting with suspicion of giant cell arteritis—lessons from a vasculitis clinic. Eur Heart J - Cardiovasc Imaging. 2018;19:933-40

3) Berger CT, Sommer G, Aschwanden M, Staub D, Rottenburger C, Daikeler T. The clinical benefit of imaging in the diagnosis and treatment of giant cell arteritis. Swiss Med Wkly [Internet]. 22. August 2018

Reviewer 3 Report

Very nice and consise manuscript. The equation to define GCA comes to a simple definition which is great after all attempts in the field.

Only two remarks:

  • The injected activity (5 MBq/kg) is quite high. How the authors justify this? Would they discuss it in the Discussion section? But, in other words, do they think similar results can be achieeved with lower activities as currently proposed (2.5-3.0 MBq/kg)
  • In the ROC analysis, and in particular in Fig. 2, I would suggest to add the s.d. (or variance)of the AUC, to make it more scientific.

Author Response

Reviewer 3

 Reviewer 3 Very nice and concise manuscript. The equation to define GCA comes to a simple definition which is great after all attempts in the field.

Only two remarks:

Point 1: The injected activity (5 MBq/kg) is quite high. How the authors justify this? Would they discuss it in the Discussion section? But, in other words, do they think similar results can be achieved with lower activities as currently proposed (2.5-3.0 MBq/kg)

>Point 1: Indeed, the European Guideline [1] recommends activities of 2–3 MBq/kg 18F-FDG for FDG-PET/CT imaging in large vessel vasculitis, combined with a scan duration of 2-3 min/bed position, depending on vendor suggestion of camera system.

However, national guidelines by the German association of Nuclear Medicine [2,3], endorsed by the Swiss association of Nuclear Medicine, recommend higher activities for PET/CT imaging (350 MBq/70 kg, corresponding to 5 MBq/kg). In the guideline, these activities are justified in order to keep acquisition time short, aiming to optimization of PET/CT co-registration. This allowed us to scan patients with a scan duration of 90 sec/bed position in the past, including the interval of the study.

After the installation of more sensitive PET/CT scanners, we were able to scan after injection of 3.5 MBq/kg with a scan duration of 90 sec/bed position since 11/2021, which was after the acquisition of the patient scans for our study.

However, we do not expect different results of the method described in our study after injection of lower activities than 5 MBq/kg, as SUV calculation takes into account injected activities and, furthermore, our method normalizes SUV values by the use of a ratio.

1) Slart, R.H.J.A., Writing group, Reviewer group. et al. FDG-PET/CT(A) imaging in large vessel vasculitis and polymyalgia rheumatica: joint procedural recommendation of the EANM, SNMMI, and the PET Interest Group (PIG), and endorsed by the ASNC. Eur J Nucl Med Mol Imaging 45, 1250–1269 (2018).

2) DGN-Handlungsempfehlung (S1-Leitlinie). Differentialindikation für verschiedene radioaktive Arzneimittel bei unterschiedlichen entzündlichen Erkrankungen. AWMF-Registernummer: 031-018

3) Krause BJ, Beyer T, Bockisch A, Delbeke D, Kotzerke J, Minkov V, Reiser M, Willich N. FDG-PET/CT in der Onkologie. Leitlinie [FDG-PET/CT in oncology. German Guideline]. Nuklearmedizin. 2007;46(6):291-301.

Point 2: In the ROC analysis, and in particular in Fig. 2, I would suggest to add the s.d. (or variance) of the AUC, to make it more scientific.

>Point 2:  We now added the SE values in the results section.

Round 2

Reviewer 2 Report

Although the reviewer understood their claims, the information regarding point 1 to 3 was not added in the manuscript. The reviewer recommends to revise the corresponding parts in the manuscript.

Author Response

Point 1: Although the reviewer understood their claims, the information regarding point 1 to 3 was not added in the manuscript. The reviewer recommends to revise the corresponding parts in the manuscript.

>Point 1: We have adapted the regarding information accordingly.

Round 3

Reviewer 2 Report

The authors revised manuscript adequately. 

Author Response

Thank you